# Improving atmospheric path-attenuation estimates for radiopropagation applications by microwave radiometric profiling

Ayham Alyosef[1], Domenico Cimini[2,1], Lorenzo Luini[3], Carlo Riva[3], Frank S. Marzano[4,1], Marianna Biscarini[4,1], Luca Milani[5], Antonio Martellucci[5], Sabrina Gentile[2,1], Saverio T. Nilo[2], Francesco Di Paola[2], Ayman Alkhateeb[6], Filomena Romano[2]

[1] CETEMPS, University of L'Aquila, L'Aquila, 67100, Italy
[2] CNR-IMAA, C.da S.Loja, Potenza, 85100, Italy
[3] DEIB - IEIIT – CNR, Politecnico di Milano, Milano, 20100, Italy
[4] DIET, Sapienza University di Roma, Rome, 00185, Italy
[5] ESA, ESTEC/ESOC
[6] University of Aleppo, Aleppo, Syria

Correspondence to: Domenico Cimini (domenico.cimini@imaa.cnr.it)

**Abstract.** Ground-based microwave radiometer (MWR) observations of downwelling brightness temperature ($T_B$) are commonly used to estimate the atmospheric attenuation at relative transparent channels for radiopropagation and telecommunication purposes. The atmospheric attenuation is derived from $T_B$ by inverting the radiative transfer equation with a priori knowledge of the mean radiating temperature ($T_{MR}$). $T_{MR}$ is usually estimated by either time-variant site climatology (e.g., monthly average computed from atmospheric thermodynamical profiles) or condition-variant estimation from surface meteorological sensors. However, information on $T_{MR}$ may also be extracted directly from MWR measurements at other channels than those used to estimate atmospheric attenuation. This paper proposes a novel approach to estimate $T_{MR}$ in clear and cloudy sky from independent MWR profiler measurements. A linear regression algorithm is trained with a simulated dataset obtained by processing one year of radiosonde observations of atmospheric thermodynamic profiles. The algorithm is trained to estimate $T_{MR}$ at K-, and V/W-band frequencies (22-31 and 72-82 GHz, respectively) from independent MWR observations at V-band (54-58 GHz). The retrieval coefficients are then applied to a one-year dataset of real V-band observations, and the estimated $T_{MR}$ at K- and V/W-band are compared with estimates from nearly collocated and simultaneous radiosondes. The proposed method provides $T_{MR}$ estimates in better agreement with radiosondes than a traditional method, with 32-38% improvement depending on frequency. This maps into an expected improvement in atmospheric attenuation of 10-20% for K-band and ~30% for V/W-band channels.

## 1 Introduction

There is a continuous trend to use higher frequencies in the development of Satellite Communication (SatCom), as lower frequency bands become saturated (e.g., Biscarini et al., 2017). Europe's current Earth observation programs with the Sentinel satellite constellation generate a daily data volume of terabytes, requiring new broadband links to access the data. In future interplanetary explorer missions, the need for high throughput communications will also become more pressing due to a wider range of observed parameters and teleoperated landers or rovers, avoiding data loss due to limited on-board memory or data

compression (Jebril et al., 2007; Acosta et al., 2012). In remote areas on Earth, like Antarctica, it is of concern to forward scientific data via satellite to the research facilities (Bonifazi et al., 2002). All mentioned scientific applications have in common that the increase in data volume requires higher transmission capacities than those available nowadays. Current high-throughput SatCom systems operate at X (8–12 GHz), Ku (12-18 GHz), K (18-26) and Ka (26-40 GHz) band and presumably their next implementation will use Q (40-50 GHz) and V (50-75 GHz) bands, whereas W band (75-110 GHz) appears to be

the next natural evolution (Riva et al., 2014). Moving beyond the X and Ku bands to less congested higher frequencies increases the available bandwidth, allowing smaller equipment that consequently reduce the size of the satellite and launch vehicle (Cianca et al., 2011; Acosta et al., 2012; Emrick et al., 2014).

       Ground-based microwave radiometer (MWR) observations of downwelling brightness temperature ($T_B$) are commonly used to estimate the atmospheric attenuation at relatively transparent microwave channels for radiopropagation and

telecommunication purposes (e.g., Marzano et al., 2006; Marzano, 2007; Biscarini et al., 2019). However, higher frequencies are characterized by larger dynamics of atmospheric propagation effects, mainly because of higher atmospheric losses (rain, clouds and atmospheric gases). Planning of V and W band SatCom systems require experimental data to characterize these unexplored atmospheric radio channels (Mattioli et al., 2013; Riva et al., 2014; Biscarini and Marzano, 2020). Radiowave propagation models can provide a reliable estimate of the atmospheric path attenuation, but have been typically validated only

for frequencies up to 50 GHz (Riva et al., 2014). These models, recommended by the International Telecommunication Union (ITU), are based on past experimental campaigns at K/Ka and Q bands, whereas designing the Earth-satellite link budget at V and W bands would require satellite beacon data which are currently not available. It is then essential to investigate the behaviour of electromagnetic waves in the V and W bands to improve existing models and validate them by independent measurements (Lucente et al., 2008; Biscarini et al., 2019).

In response to this need, a measurement campaign has been recently planned to characterize the V- and W-band satellite atmospheric radio channel through ground-based microwave radiometric observations. The core observatory is located at Politecnico di Milano (Milan, Italy), where a four-channel MWR, including two V and W band channels at 72.5 and 82.5 GHz, respectively, are operated. An independent MWR, a 14-channel temperature and humidity profiler, is also operated in Spino d'Adda, 25 km from Milan (Italy). Atmospheric path attenuation is derived from MWR $T_B$ observations inverting the radiative

transfer equation with a prior knowledge of the mean radiating temperature ($T_{MR}$). A priori $T_{MR}$ is usually obtained either by monthly average values computed from radiosondes (e.g., Martellucci, 2007) or inferred from surface meteorological sensors

(e.g., Luini et al., 2018) or derived from radiopropagation models (e.g., Mattioli et al., 2013; Biscarini and Marzano, 2020). The uncertainty in $T_{MR}$ estimates contributes to the path attenuation uncertainty. To the aim of reducing this uncertainty, in this work we propose an original approach increasing the accuracy of $T_{MR}$ estimates by exploiting independent MWR profiler measurements. This is a follow up of the work presented at the 11th International Symposium on Tropospheric Profiling (Cimini et al., 2019). The paper is structured as follows: Section 2 describes the methodology, whereas Section 3 presents the available dataset; Section 4 presents the results and the obtained performance, and Section 4 summarises the results, providing hints for future work.

## 2 Methodology

The atmospheric brightness temperature $T_B$ (K), measured by a MWR at frequency $f_i$ and elevation angle $\theta$, can be used to estimate the atmospheric total path attenuation $A_{MWR}(f_i, \theta)$ (in dB) using the following expression (e.g., Marzano, 2007; Ulaby et al., 2014):

$$A_{MWR}(f_i, \theta) = 10 log_{10}\left(\frac{T_{MR}(f_i, \theta) - T_C}{T_{MR}(f_i, \theta) - T_B(f_i, \theta)}\right) \qquad (1)$$

where $T_C$ is the cosmic background temperature (usually set to 2.73 K in the microwave and millimeter-wave range) and $T_{MR}(f_i, \theta)$ is the mean radiating temperature (in K), which is given by (e.g., Han and Westwater, 2000):

$$T_{MR} = \frac{\int_0^\infty T(s)\alpha(s)e^{-\tau(0,s)}ds}{\int_0^\infty \alpha(s)e^{-\tau(0,s)}ds} \qquad (2)$$

where $T(s)$ and $\alpha(s)$ are the atmospheric physical temperature and absorption coefficient along the path $s$ and $\tau(0, \infty) = \int_0^\infty \alpha(s)ds$ is the total atmospheric opacity (Np) from surface to the top of the atmosphere. As Eq.(2) suggests, the mean radiating temperature represents the mean temperature along the optical path weighted by the atmospheric transmission $T_A = e^{-\tau}$, i.e. the inverse of the atmospheric loss $L_A = e^{\tau}$. Note that Eq.(1) and (2) are derived from the radiative transfer equation for a non-scattering atmosphere (Swarztchild's equation) and adopting the Rayleigh-Jeans approximation (Janssen, 1993), commonly used in the microwave range to simplify the Planck's law with a linear relationship with temperature $T$, $B_f(T) \approx 2k\frac{f^2}{c^2}T$, where $k$ and $c$ are the Boltzmann and speed of light constants, respectively. In these conditions, the atmospheric opacity can be written as:

$$\tau = ln(\frac{T_{MR}(f_i,\theta)-T_C}{T_{MR}(f_i,\theta)-T_B(f_i,\theta)}) \qquad (3)$$

and thus the atmospheric total path attenuation, which is simply the atmospheric loss in dB units, can be rewritten in terms of $\tau$ as:

$$A_{MWR} = 10log_{10}(e^{\tau}) = \frac{10}{ln10}ln(e^{\tau}) = \frac{10}{ln10}\tau = 4.343\,\tau \qquad (4)$$

Note that, as discussed in Han and Westwater (2000) and Janssen (1993), Eq.(3) is just an approximation of the exact formulation. In the frequency range used here, this approximation is valid within 2% from the exact formulation, and thus it is
adopted here for the sake of simplicity. Moreover, atmospheric scenarios with rainfall and snowfall are excluded since multiple scattering is not included in Eq.(1) and thus in this work (see Marzano et al., 2006; Biscarini and Marzano, 2019).$T_{MR}$ can be easily calculated from the atmospheric profiles of physical temperature and absorption coefficient through Eq.(2). In clear-sky conditions, radiosonde profiles of temperature and humidity are sufficient to compute $T_{MR}$, while in presence of clouds assumptions must be made on the vertical distribution of condensed water (e.g., Salonen and Uppala, 1991).

Thus, the mean radiating temperature plays a role in mapping the brightness temperature to the atmospheric opacity and then total path attenuation, and the operational estimate of atmospheric attenuation from radiometric $T_B$ observations requires some *a priori* knowledge on $T_{MR}$. Traditionally, $T_{MR}$ was treated as a constant determined climatologically from a dataset of atmospheric profiles, usually radiosondes. This assumption propagates uncertainty in the attenuation estimates through Eq.(1). However, as long as $T_B$ is relatively low, e.g., for zenith and low frequency observations, the $T_{MR}$ uncertainty contribution to
attenuation is rather small, and thus a precise knowledge of $T_{MR}$ is not crucial.

On the other hand, with increasing $T_B$ values, e.g. in case of observations at lower elevation angles and/or at relatively more opaque higher frequencies, accurate $T_{MR}$ estimates gain more importance. One consequence is that $T_{MR}$ uncertainties cause significant calibration errors when large air masses (i.e. pointing at low elevation angle) are used. For example, it has been demonstrated that using a $T_{MR}$ climatological mean (with 9 K standard deviation, based on a 13-year dataset) introduces up to
1.4 K uncertainty in tipping curve calibration at K-band channels, exploiting elevation angles down to ~15° (Han and Westwater, 2000).

Thus, methods are usually exploited to reduce $T_{MR}$ uncertainties, especially when low angle and/or high frequency observations are involved. One simple method is to divide the $T_{MR}$ climatology into seasons, efficiently reducing the standard deviation of the climatological mean. A slightly more sophisticated method exploits time interpolation of the $T_{MR}$ monthly mean
(Martellucci, 2007). However, these methods do not consider the actual meteorological conditions, which may significantly differ from the seasonal or monthly mean. In order to consider the actual meteorological conditions, another method is the predicting of $T_{MR}$ from the surface air temperature, using regression analysis. Surface-based temperature measurements along with $T_{MR}$ calculated from radiosonde measurements provide the means to derive linear regression coefficients relating surface temperature to $T_{MR}$. It has been shown that this method reduces the calibration uncertainty in K-band channels by a factor of

~3 (Han and Westwater, 2000). Other surface measurements, such as pressure and humidity may also be considered among the predictors in addition to temperature. This last method, relating $T_{MR}$ to surface pressure, temperature, and humidity (PTU) measurements, likely represents the current best practices (Luini et al., 2018). Note that hereafter relative humidity is used as the humidity variable.

However, the PTU method may be inaccurate in particular cases, i.e. when surface conditions are not well correlated with
upper air. One obvious case is the occurrence of strong temperature inversions. To circumvent this problem, another method was suggested by Han and Westwater, (2000): $T_{MR}$ prediction could be improved by using boundary temperature profiles from a MWR profiler or a radio acoustic sounding system, which accurately recovers boundary layer surface temperature inversions (Martner et al., 1993). To our knowledge, this suggestion has not been demonstrated yet.

Thus, this analysis builds on this suggestion and presents a method to derive $T_{MR}$ from combined surface measurements and
MWR profiler observations, demonstrating the reduced uncertainty with respect to the other methods introduced above.

## 3 Dataset and implementation

The proposed method is demonstrated estimating $T_{MR}$ at four channels in K- and V/W-bands from surface measurements and independent MWR profiler observations. The data set considered here consists of experimental data collected in 2015-2016 at
two sites involved within the ESA WRad campaign. The MWR operated in Spino d'Adda is a Humidity and Temperature profiler (HATPRO) manufactured by Radiometer Physics GmbH (RPG), measuring $T_B$ at 14 channels from K- to V-band (22.24, 23.04, 23.84, 25.44, 26.24, 27.84, 31.4, 51.26, 52.28, 53.86, 54.94, 56.66, 57.3, 58.0 GHz). The MWR operated at Politecnico di Milano is a LWP-U72-82 manufactured by RPG, measuring $T_B$ at 4 channels, two K-band (23.84 and 31.4 GHz), and two between V- and W-bands (72.5 and 82.5 GHz). During the considered period, both MWR pointed constantly at ~35°
elevation towards the geostationary satellite Alphasat, collecting one sample per second. Standard meteorological sensors are located near the two MWR to provide the environmental PTU measurements.

In addition, the dataset includes the atmospheric thermodynamical profiles measured by radiosondes launched operationally twice a day from the Linate airport in Milan (~5 km from Politecnico di Milano). The two radiosondes per day are launched at 11.30 and 23.30 UTC. Radiosonde profiles in the period from January 2015 to December 2016 have been collected for this
analysis. Atmospheric thermodynamical profiles from each radiosonde have been processed to compute the simulated $T_{MR}$ in clear and cloudy conditions, using the Wave Propagation Laboratory (WPL) radiative transfer code. This code was originally developed at the U.S. National Oceanic and Atmospheric Administration (NOAA, Schroeder J. A. and E. R. Westwater, 1991), implementing the millimeter-wave propagation model (MPM, Liebe, 1989), and has since been updated with refined spectroscopic parameters (Rosenkranz, 2017), as described in Cimini et al. (2018) and references therein. The cloud water
content is modeled using the TKK method (Salonen and Uppala, 1991; Luini et al., 2018).

The experimental implementation is pictured in Figure 1. $T_B$, $T_{MR}$, and PTU simulated from the two-year dataset of radiosonde profiles are used in the training and test phases. Synthetic noise, with zero mean and standard deviation equal to the expected

instrument accuracy, has been added to simulate the instrument uncertainty. In the training phase, a half data set (2016) is used to train two versions of a multivariate linear regression to estimate $T_{MR}$ from either PTU only or PTU and $T_B$. From the set of fourteen HATPRO channels available, we selected the five higher frequency V-band channels (51.26, 52.28, 53.86, 54.94, 56.66, 57.3, 58.0 GHz). These channels are mostly sensitive to atmospheric temperature and are less affected by hydrometeors than lower frequency K-band channels, which makes them more suited for the operational whole-sky estimate of $T_{MR}$. In the test phase, the two versions of regression coefficients are used to estimate $T_{MR}$ from either PTU only or PTU and $T_B$ from the remaining dataset (2015). The resulting $T_{MR}$ are then compared with "true" values computed from simultaneous radiosondes. Finally, in the validation phase, the two versions of regression coefficients are fed with real measurements, either from PTU sensor only or with PTU sensor and HATPRO five V-band channels. The resulting $T_{MR}$ values are again compared with "true" radiosonde values, and also applied to real LWP-U72-82 observations to estimate atmospheric attenuation through Eq(1).

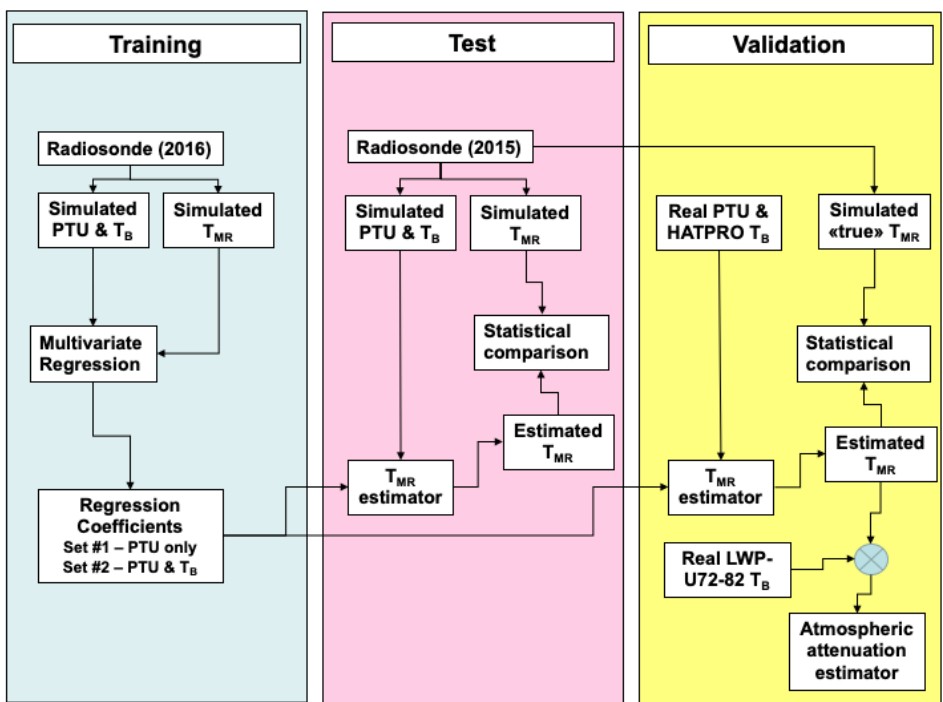

**Figure 1: Flow chart of the implemented data analysis.**

## 4 Results

In the validation phase, the multivariate regression trained with the simulated dataset from 2016 is applied to real observations in 2015 and validated against $T_{MR}$ computed from radiosonde profiles. For the considered pointing angle (35° elevation), the cloud liquid water path estimated from radiosondes reaches 2.8 mm for the training set, while the liquid water path estimated

from MWR observations within the validation set reaches 4.6 mm. The results from the two versions of regression coefficients, one applicable to surface PTU measurements only and the other applicable to PTU measurements and five V-band channels $T_B$, are compared here. The implemented equation and coefficients for the multivariate regression are given in Appendix A. The output dataset consists of $T_{MR}$ and $A$ at four frequencies (23.84, 31.4, 72.5, and 82.5 GHz) retrieved at 1-minute temporal resolution. One example of 24 h time series is shown in Figure 2. For all the 4 considered frequencies, it is evident that $T_{MR}$ from PTU and $T_{MR}$ from PTU&$T_B$ follow a similar diurnal cycle, decreasing up to 5 am, then rapidly increasing until noon, then remaining stable for a few hours, and finally decreasing again after 5 pm. However, there seems to be a factor of ~2 in the peak-to-peak variation, e.g., at 23.84 GHz, $T_{MR}$ peak-to-peak variation is ~9 K for $T_{MR}$(PTU) while ~4 K for $T_{MR}$(PTU&$T_B$). $T_{MR}$ computed from the two daily radiosondes, representing our reference "truth", seems to confirm that $T_{MR}$(PTU&$T_B$) is correct in estimating a smaller variation. The statistical comparison from the validation phase is reported in Figures 2 and 3, considering a set of 638 radiosondes in 2015. From this dataset, the $T_{MR}$ climatological variations in Milan in clear and cloudy sky is estimated to be ~7.6-8.2 K, depending upon K- and V/W-band channels. Time colocation with radiometric observations is achieved averaging the estimated $T_{MR}$ within 15 minutes from the radiosonde release time. All the considered statistical scores show that $T_{MR}$(PTU&$T_B$) agrees better than $T_{MR}$(PTU) with the reference radiosondes, for all the four considered frequency channels (two K- and two V/W-band). In particular, the average difference (AVG), the root-mean-square difference (RMS), and the correlation coefficient (COR) with respect to $T_{MR}$ from radiosondes are reported in Table 1. Four methods to estimate $T_{MR}$ are reported in Table 1: seasonal climatology (monthly mean), time-interpolated monthly mean, regression from PTU, and finally regression from PTU & $T_B$. As one would expect, Table 1 indicates that condition-dependent methods (e.g., the two regression types) outperform methods simply based on climatology. The only score being better for climatology methods is AVG, i.e. the average difference over one year. This is somewhat expected, as the climatology methods minimize the annual mean difference by definition. Nonetheless, the regression methods show modestly higher AVG values. Conversely, the regression methods show substantially better RMS and COR scores with respect to climatological methods, which confirms that regression methods are preferable when accurate estimates of $T_{MR}$ and atmospheric attenuation are desired. Table 1 also clearly indicates that the regression based on PTU & $T_B$ outperforms the one based on PTU only. For the considered K- and V/W-band frequencies, the improvement ranges between ~0.2-0.8 K in average difference, ~1.0-1.4 K in RMS, and ~4-7% in correlation. This demonstrates quantitatively that the consideration of V-band channels within the regression brings in significant information on $T_{MR}$, as originally foreseen by Han and Westwater (2000).

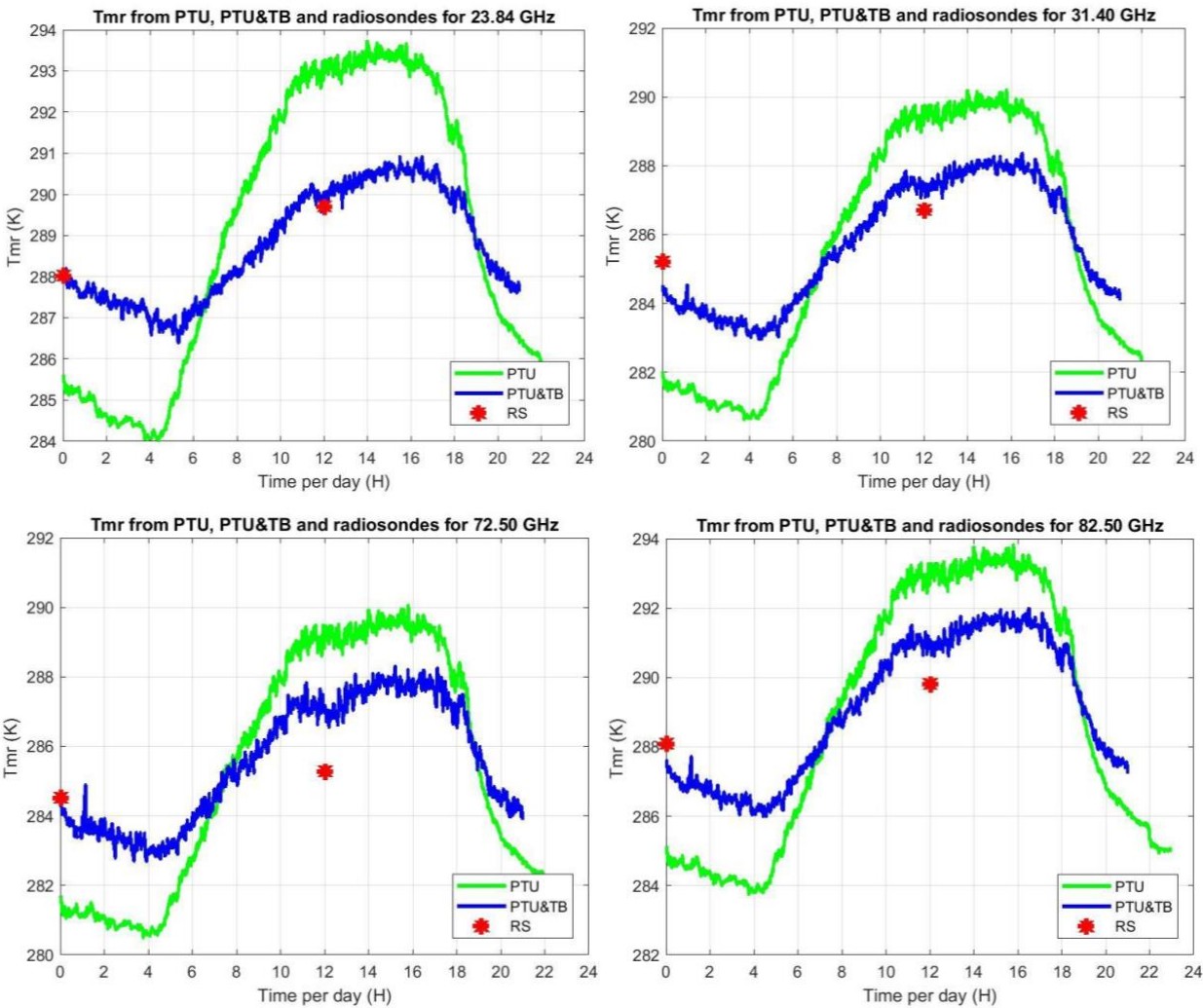

**Figure 2: 24 h time series (19 July 2015) of $T_{MR}$ as estimated from surface PTU measurements (green line) and with the additional $T_B$ at five V-band channels (blue line). $T_{MR}$ from twice-daily radiosonde measurements are also reported (red dots). Clockwise from top-left: 23.84, 31.40, 72.50, 82.50 GHz.**

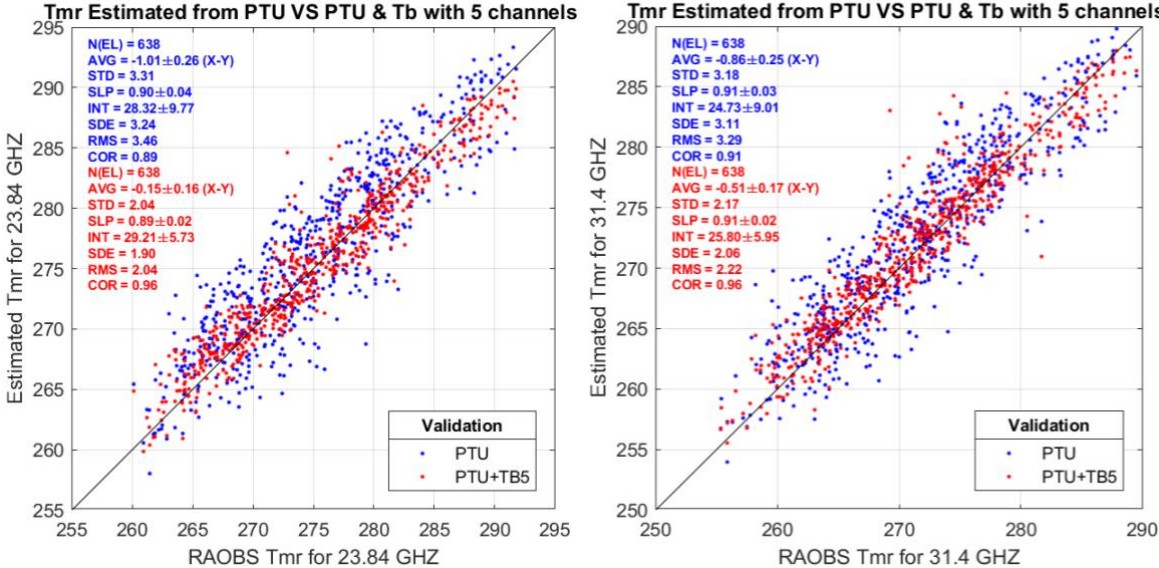

**Figure 3: Estimated vs reference $T_{MR}$ at K-band. Left: 23.84 GHz. Right: 31.4 GHz. Red dots indicate estimated $T_{MR}$ based on (PTU, $T_B$), while blue dots indicate $T_{MR}$ based on (PTU) only. Each panel reports the number of elements (N(EL)), the average difference (AVG), the standard deviation (STD), the slope (SLP) and intercept (INT) of a linear fit, the standard error (SDE), the root-mean-square (RMS), and correlation coefficient (COR). 95% confidence intervals are given for AVG, SLP, and INT. Units for AVG, STD,**

**SDE, RMS are kelvin.**

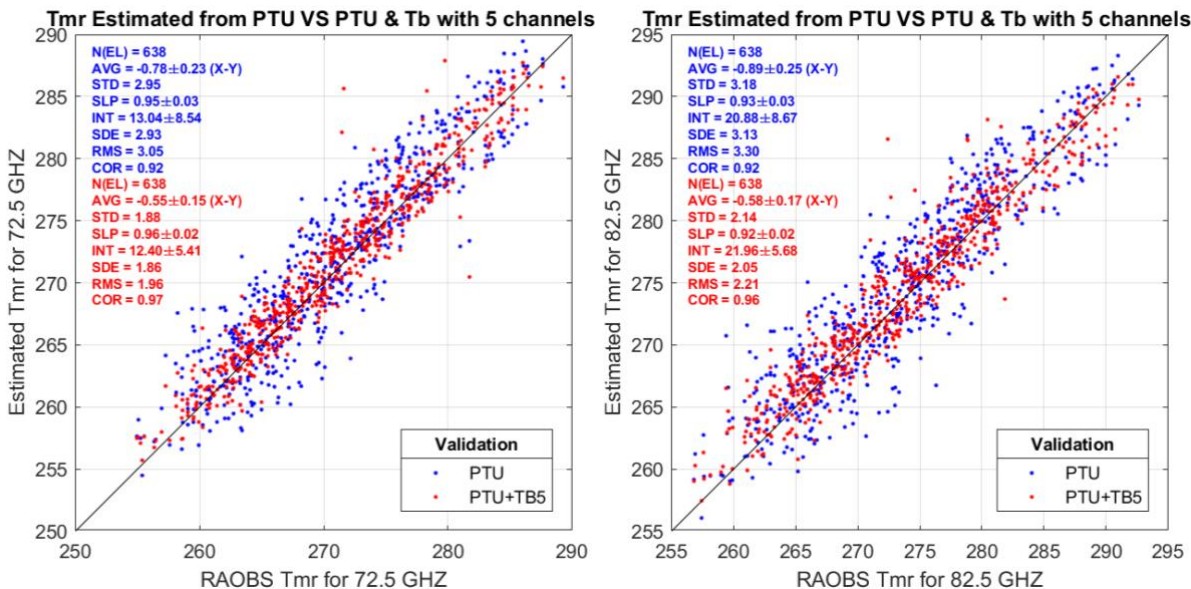

**Figure 4: As in Fig. 3 but for V- and W-bands channels. Left: 72.5 GHz. Right: 82.5 GHz.**

**Table 1: Average difference (AVG), root-mean-square difference (RMS), and correlation coefficient (COR) with respect to reference $T_{MR}$ (computed from radiosondes) for $T_{MR}$ estimated from four methods: Monthly mean, time-interpolated monthly mean, regression from PTU, and regression from PTU&$T_B$. The best scores are in bold.**

| Channel | | Monthly mean | Time-interpolated monthly mean | Regression from PTU | Regression from PTU & $T_B$ |
|---|---|---|---|---|---|
| 23.84 GHz | AVG (K) | 0.57 | 0.59 | -1.01 | **-0.15** |
| | RMS (K) | 4.02 | 3.93 | 3.46 | **2.04** |
| | COR (-) | 0.82 | 0.83 | 0.89 | **0.96** |
| 31.40 GHz | AVG (K) | 0.54 | 0.58 | -0.86 | **-0.51** |
| | RMS (K) | 4.06 | 3.95 | 3.29 | **2.22** |
| | COR (-) | 0.83 | 0.85 | 0.91 | **0.96** |
| 72.50 GHz | AVG (K) | **0.40** | **0.44** | -0.78 | -0.55 |
| | RMS (K) | 3.75 | 3.61 | 3.05 | **1.96** |
| | COR (-) | 0.85 | 0.86 | 0.92 | **0.97** |
| 82.50 GHz | AVG (K) | **0.51** | **0.55** | -0.89 | -0.58 |
| | RMS (K) | 4.20 | 4.08 | 3.30 | **2.21** |
| | COR (-) | 0.83 | 0.85 | 0.92 | **0.96** |

Given the radio propagation purposes, the question is whether the improvements in $T_{MR}$ estimation given in Table 1 bring

significant improvements in atmospheric attenuation estimates. In order to investigate this, we propagate $T_{MR}$ and $T_B$ uncertainty through Eq(1) to obtain the uncertainty of atmospheric attenuation. From Eq.(3-4), the uncertainty in atmospheric attenuation is simply related to the uncertainty in atmospheric opacity as:

$$\sigma_A = 4.343 \; \sigma_\tau \qquad (5)$$

where

$$\sigma_\tau = \left[\left(\frac{T_C-T_B}{(T_{MR}-T_C)(T_{MR}-T_B)}\right)^2 \sigma_{TMR}{}^2 + \left(\frac{1}{(T_{MR}-T_B)}\right)^2 \sigma_{TB}{}^2\right]^{1/2} \quad (6)$$

is the uncertainty in atmospheric opacity due to the uncertainty in $T_{MR}$ and $T_B$, respectively $\sigma_{TMR}$ and $\sigma_{TB}$. Thus, we compute the uncertainty of atmospheric attenuation $\sigma_A$ in case $T_{MR}$ is estimated from PTU&$T_B$ and from PTU only, by replacing in Eq(6) $\sigma_{TMR}$ with the $T_{MR}$ uncertainty in Table 1 and $\sigma_{TB}$ with a typical value for MWR $T_B$ uncertainty, i.e. 0.5 K (e.g., Cimini et al., 2003). The percentual improvement brought by the $T_{MR}$ estimated with the proposed method (#2, based on PTU&$T_B$) over the conventional method (#1, based on PTU only) is quantified by

$$I = \frac{\sigma(\#1) - \sigma(\#2)}{\sigma(\#1)} * 100 \quad (7)$$

both for $T_{MR}$ and $A$. Table 2 summarizes the percentual improvements for the four considered frequencies in the K- and W-band. Thus, with respect to conventional PTU method, the proposed method in average improves the $T_{MR}$ estimates by more than 32% and it is expected to improve the $A$ estimates by 10-20% at K-band channels and ~30% at V/W bands channels. In terms of radio propagation measurements, the achieved improvement level is rather modest (fraction of a dB) in clear-sky conditions, when $T_B$ and the atmospheric attenuation are low, but becomes more and more important as $T_B$ and the attenuation increase (e.g., heavy clouds and precipitation), due to the $(T_{MR} - T_B)$ factor at the denominator of Eq.(1) and (6).

Table 2: Percentage improvements brought by the proposed method (based on PTU & $T_B$) over the conventional method (based on PTU only). Note that while the improvements for $T_{MR}$ are validated against radiosondes (i.e. the STD in Figures 3-4), the improvements for $A$ are estimated through Eq(5-6), and thus represent an estimate of the expected improvements.

| Channel frequency (GHz) | 23.84 | 31.40 | 72.50 | 82.50 |
|---|---|---|---|---|
| $\sigma_{TMR}$ (K) for PTU method | 3.31 | 3.18 | 2.95 | 3.18 |
| $\sigma_{TMR}$ (K) for PTU&$T_B$ method | 2.04 | 2.17 | 1.88 | 2.14 |
| $T_{MR}$ uncertainty improvement (%) | 38 | 32 | 36 | 33 |
| $A$ uncertainty improvement (%) | 24 | 14 | 32 | 28 |

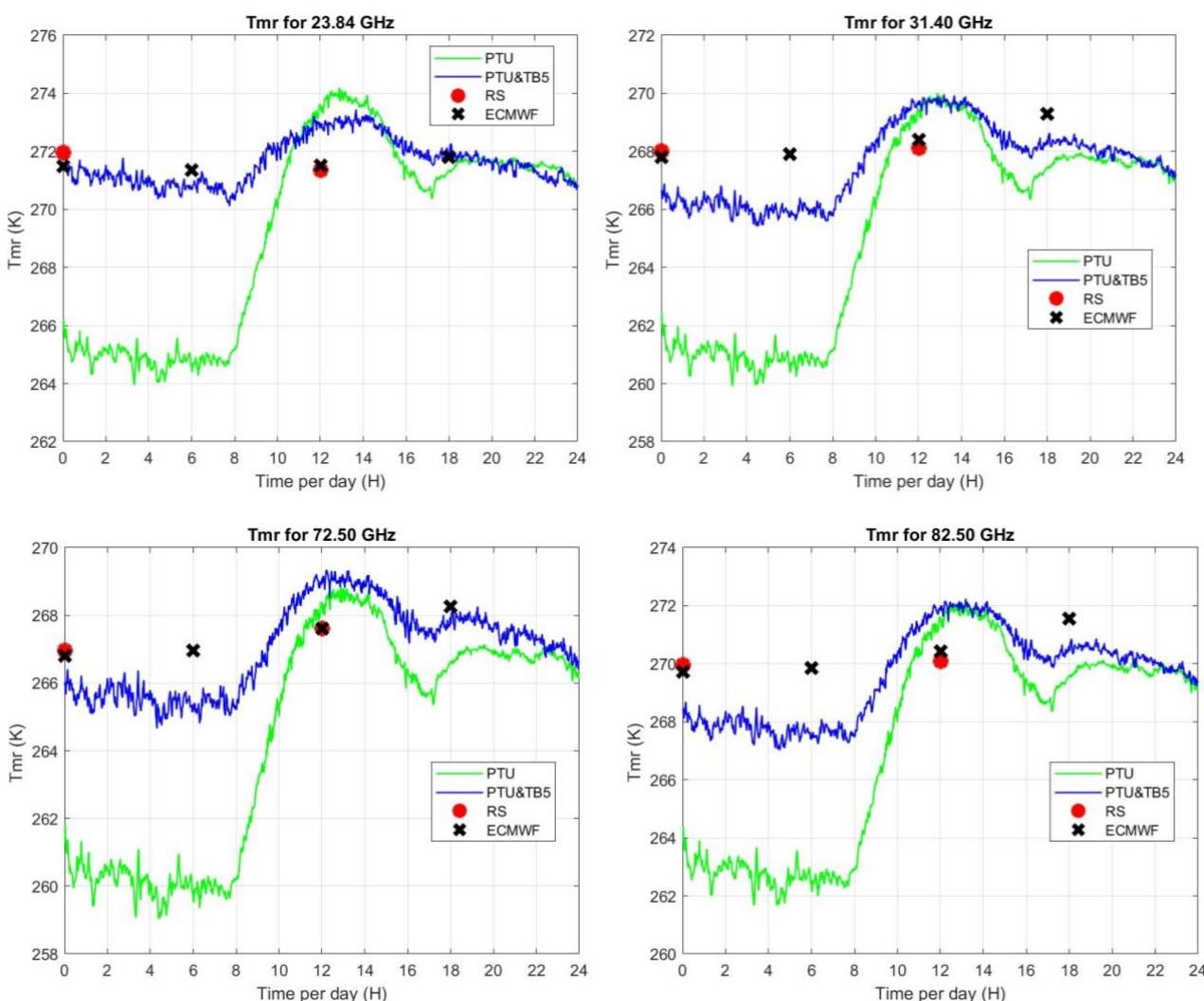

**Figure 5: 24 h time series (31 Dec 2018) of $T_{MR}$ as estimated from surface PTU measurements (green line) and with the additional $T_B$ at five V-band channels (blue line). $T_{MR}$ from twice-daily radiosonde measurements (red dots) and from ECMWF analysis (black crosses) are also reported. Clockwise from top-left: 23.84, 31.40, 72.50, 82.50 GHz.**

To show an example of application, we select one day (31 December 2018) for which data from the 14-channel MWR in Spino d'Adda and the four-channel MWR at Politecnico di Milano are both available, together with the PTU readings. PTU and $T_B$ at the five higher frequency V-band channels (51.26, 52.28, 53.86, 54.94, 56.66, 57.3, 58.0 GHz) of the 14-channel MWR are used to compute $T_{MR}$ at the frequencies of the four-channel MWR (23.84, 31.40, 72.50, 82.50 GHz). $T_{MR}$ and the observed $T_B$ at the four channels are used to compute the attenuation. Results for both PTU and PTU&$T_B$ methods are shown in Figure 5

($T_{MR}$) and Figure 6 (attenuation). Figures 5 and 6 also show $T_{MR}$ and attenuation computed from the radiosonde profiles (twice daily) and the model profiles (every six hours) from the nearest grid point of global analysis produced by the European Centre

for Medium-range Weather Forecast (ECMWF). The difference between PTU and PTU&$T_B$ methods is evident between midnight and 8 am. As indicated by the radiosonde profile (not shown), that night was characterised by a temperature inversion near the surface, about 8 K strong and 160 m deep. This causes the surface temperature (used in the PTU method) to decouple from that of upper air. Conversely, the PTU&$T_B$ method brings in information on lower atmospheric temperature. $T_{MR}$ difference between the two methods is 4-6 K at 8 am, rapidly decreasing as the Sun warms up the surface, and fading to negligible values around noon.

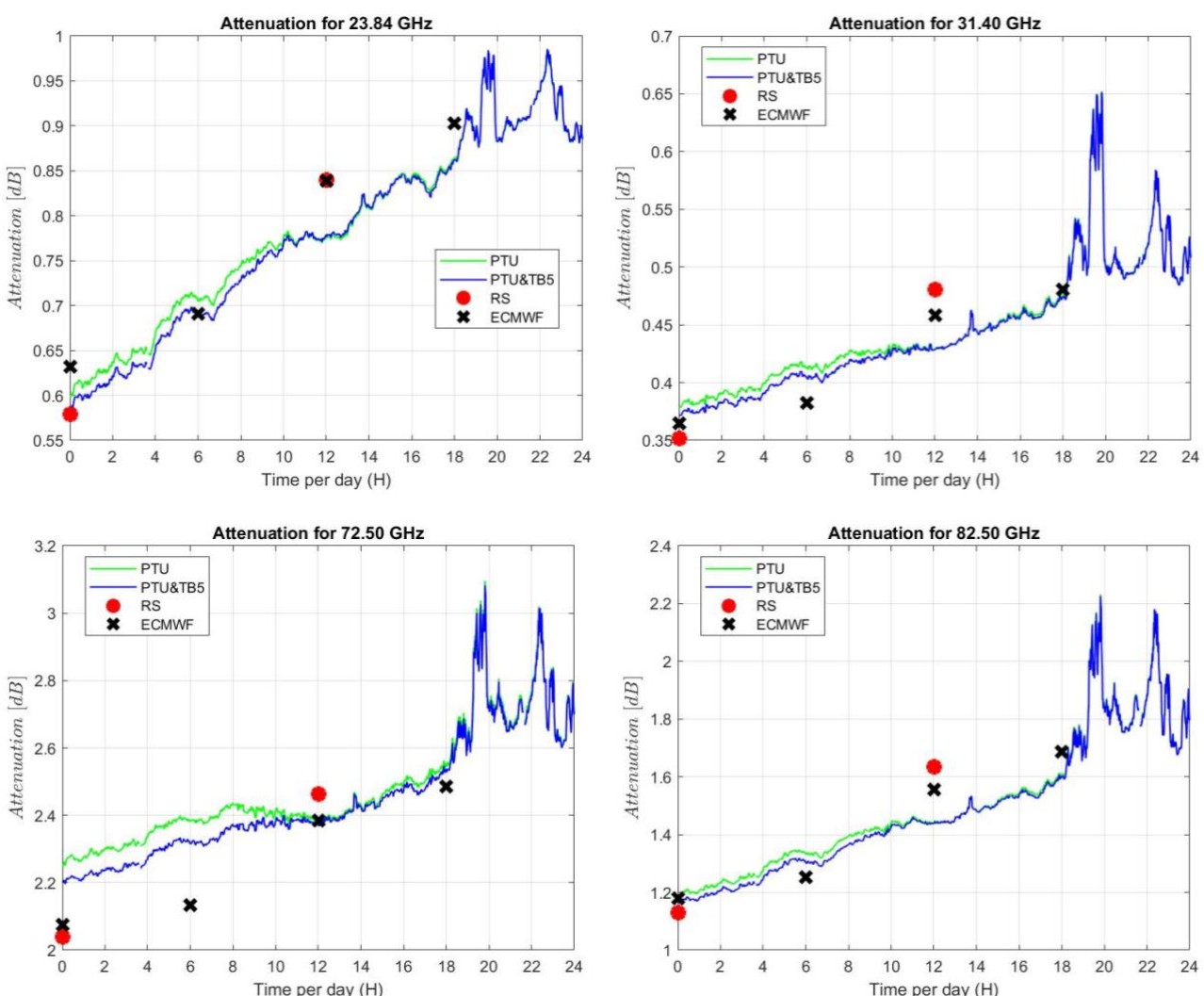

Figure 6: 24 h time series (31 Dec 2018) of $A$ from PTU (green line) and PTU&$T_B$ (blue line) methods. $A$ computed from twice-daily radiosonde measurements (red dots) and ECMWF analysis (black crosses) are also reported. Clockwise from top-left: 23.84, 31.40, 72.50, 82.50 GHz.

A similar behaviour is found in attenuation (Figure 6), although the difference is less striking. Attenuation from radiosondes
and ECMWF profiles are mostly closer to that from PTU&$T_B$ method. However, a proper validation would require a data set
with larger dynamical range, and an independent reference valid in both clear and cloudy conditions. In fact, neither radiosonde
nor ECMWF profiles can be assumed as reference in cloudy conditions for the lack of accurate cloud water content, which for
radiosondes is modelled statistically (TKK method), while for ECMWF it represents a larger scale than the local one. The
collection of a reference dataset is indeed the main objective of the WRad campaign, through the application of Sun-tracking
microwave radiometry (Biscarini et al., 2019 and references therein).

# 5 Conclusions

In this paper we propose an approach to estimate $T_{MR}$ from radiometric observations at V-band (sensitive to atmospheric
temperature) in addition to surface measurements of PTU, which represents the current best practice. The approach was
suggested in Han and Westwater (2000) but never attempted to our knowledge. Here, we implement the suggested approach
by applying multivariate linear regression to radiometric and radiosonde observations collected in the Milan area (Italy). Two
independent microwave radiometers are considered, one atmospheric profiler operating at 14 channels in the K- and V-bands,
and one 4-channel radiometer operating at two K- and two between V- and W-bands channels. The implemented approach
exploits five V-band channels of the microwave profiler (namely at 53.86, 54.94, 56.66, 57.3, and 58.0 GHz) together with
surface PTU measurements to estimate $T_{MR}$ at the K- and V/W-band frequencies of the four-channel radiometer. The
conventional method is also implemented, estimating $T_{MR}$ at the frequencies of the four-channel radiometer from PTU
measurements only. Results from the proposed and conventional methods are validated against $T_{MR}$ from simultaneous
radiosondes, showing improvement in all channels and statistical scores (~0.2-0.8 K in average difference, ~1.0-1.4 K in RMS,
and ~4-7% in correlation, depending upon frequency). This corresponds to a decrease in $T_{MR}$ estimation uncertainty by 32 to
38%, depending upon frequency. The improvement in $T_{MR}$ estimation is then mapped into the improvement in attenuation
estimates for radio propagation purposes by propagating typical $T_{MR}$ and $T_B$ uncertainties into the atmospheric attenuation
equation. This results in expected improvements in atmospheric attenuation estimates of the order of 10-20% at K-band
channels and ~30% at V/W-band channels. Although this level of improvement leads to modest change in absolute attenuation
in clear sky (fraction of a dB), it becomes more and more important (few dBs) with the increasing attenuation typical of cloudy
and rainy conditions. In summary, this paper demonstrates the validity of Han and Westwater (2000)'s idea, and it provides
quantitative assessment of the improvements brought by the proposed method over the conventional PTU method for
estimating $T_{MR}$ and atmospheric attenuation, at the cost of higher observation complexity (two radiometers in a relatively small
area). This limitation may be overcome by the increasing availability of MWR profilers, currently deployed at several ground
stations serving satellite telecommunication (e.g. ESA Tracking Network in Cebreros, Malargue, and New Norcia) as well as
observatories devoted to atmospheric research and operational weather forecast (Cimini et al., 2020). Concerning the radio

propagation purposes, future work will include the application of the proposed method to the dataset collected within the ESA WRad campaign (Aug. 2019 - Aug. 2021) to further validate the improvements in the atmospheric attenuation estimates in whole sky conditions, eventually contributing to the future assessment of V/W-band link budget for Earth-satellite telecommunication.

## Appendix A - Coefficients for multivariate multiple linear regression

Multivariate multiple linear regression (Bevington and Robinson, 2003) is used here to estimate $T_{MR}$ at four frequencies (23.8, 31.4, 72.5, 82.5 GHz). To clarify, note that the term multivariate refers to statistical models that have more than one dependent or outcome variable (predictands), while multiple (or multivariable) refers to statistical models that have more than one independent or input variable (predictors) (e.g., Hidalgo & Goodman, 2013). Following Cimini et al. (2006), and references therein, a general equation for the multivariate multiple linear regression between $\hat{\mathbf{x}}$ (vector of predictands) and $\mathbf{y}$ (vector of predictors) is:

$$\hat{\mathbf{x}} = \mathbf{x_0} + \mathbf{D}(\mathbf{y} - \mathbf{y_0}) \qquad (A1)$$

$$\mathbf{D} = \mathbf{C_{xy}} \mathbf{C_{yy}^{-1}} \qquad (A2)$$

where $\mathbf{D}$ is the matrix of linear regression coefficients, and $\mathbf{x_0}$, $\mathbf{y_0}$, $\mathbf{C_{xy}}$ and $\mathbf{C_{yy}}$ are estimated from the training set (*a priori* knowledge) respectively as the mean values for $\mathbf{x}$ and $\mathbf{y}$, the covariance matrix of simultaneous $\mathbf{x}$ and $\mathbf{y}$, and the autocovariance matrix of $\mathbf{y}$. In this work, the predictands $\hat{\mathbf{x}}$ are $T_{MR}$ at four frequencies. Thus, for any measured $k$-dimension vector of predictors $\mathbf{y_i}$, the estimated $T_{MR}$ for each channel $j$ is:

$$\hat{T}_{MR_i}(j) = x_0(j) + \sum_{l=1}^{k} D_{l,j} \left( y_i(l) - y_0(l) \right) \qquad (A3)$$

In this study, two versions are implemented with different sets of predictors. The first version considers three variables as predictors ($k=3$): air pressure, temperature, and relative humidity (PTU) measured by standard meteorological sensors. The second version considers eight variables as predictors ($k=8$): the three PTU readings and in addition $T_B$ at five V-band channels (53.86, 54.94, 56.66, 57.3, 58.0 GHz). From the training set, we obtain the following values for $\mathbf{x_0}$, indicating the mean $T_{MR}$ (K) at 4 frequencies:

$$\mathbf{x_0} = [275.67 \ \ 272.01 \ \ 271.66 \ \ 274.60] \qquad (K)$$

While $\mathbf{x_0}$ is the same for the two versions of multivariate multiple linear regression, both $\mathbf{y_0}$ and $\mathbf{D}$ depend on the number of predictors. For the first version $\mathbf{y}$ contains the mean PTU measurements, a vector of 3 components, and $\mathbf{D}$ is as in Table A.1:

$$\mathbf{y_0} = [1003 \ \ 288.82 \ \ 0.71] \qquad (\text{mb, K, \%/100}).$$

For the second version, $\mathbf{y}$ contains the PTU measurements and $T_B$ at five V-band channels, i.e. a vector of 8 components, and $\mathbf{D}$ is as in Table A.2:

$$\mathbf{y_0} = [276.85 \ \ 284.71 \ \ 287.07 \ \ 287.13 \ \ 287.02 \ \ 1003 \ \ 288.82 \ \ 0.71] \qquad (\text{K, K, K, K, K, mb, K, \%/100}).$$

**Table A1: D for multivariate multiple linear regression Eq.(A1-A2) to estimate $T_{MR}$ from PTU only. First row and column indicate respectively the corresponding frequency channel and predictor.**

| Frequency (GHz) | 23.8 | 31.4 | 72.5 | 82.5 | Predictor (Units) |
|---|---|---|---|---|---|
| | 0.145 | 0.140 | 0.098 | 0.128 | P (mb) |
| | 0.946 | 0.986 | 1.018 | 1.050 | T (K) |
| | 12.021 | 14.862 | 17.656 | 16.786 | RH (%/100) |

**Table A2: D for multivariate multiple linear regression Eq.(A1-A2) to estimate $T_{MR}$ from PTU & $T_B$ at five V-band channels. First row and column indicate respectively the corresponding frequency channel and predictor.**

| Frequency (GHz) | 23.8 | 31.4 | 72.5 | 82.5 | Predictor (Units) |
|---|---|---|---|---|---|
| | 0.403 | 0.690 | 1.173 | 0.810 | $T_{B\ 53GHz}$ (K) |
| | 0.555 | 0.258 | -0.273 | 0.083 | $T_{B\ 54GHz}$ (K) |
| | 0.195 | -0.082 | -0.280 | -0.111 | $T_{B\ 56GHz}$ (K) |
| | -0.140 | -0.146 | -0.037 | -0.134 | $T_{B\ 57GHz}$ (K) |
| | -0.268 | -0.150 | 0.002 | -0.095 | $T_{B\ 58GHz}$ (K) |
| | 0.066 | 0.052 | -0.013 | 0.036 | P (mb) |
| | 0.286 | 0.491 | 0.508 | 0.569 | T (K) |
| | 4.412 | 6.899 | 7.863 | 8.584 | RH (%/100) |

## Acknowledgements

This research was funded by the European Space Agency (ESA) as part of the WRad project (ESA Contract No. 4000125141/18/NL/AF). Support from COST - European Cooperation in Science and Technology (www.cost.eu) - Action CA18235 "PROBE" is also acknowledged.

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
