# Peer review of "Improving atmospheric path-attenuation estimates for"

_Atmospheric Measurement Techniques, 2020_

## Short Comment (SC1) · 21 Dec 2020

The paper is well written and explained. The analyses justifying the results are properly mentioned and explained. However, Appendix A needs to be written more comprehensively. Now it is just a bare minimum explanation. Also the mathematical formulation of the equation used in the appendix are not that clear. Readers will have difficulty in understanding the equations and the way they are written. The relationship between TMR and D is not very clear. This needs to be addressed.

---

## Referee Comment (RC1) · Ed R. Westwater (Referee) · 19 Jan 2021

The paper on improving attenuation estimates by microwave radiometry is clearly written, of scientific significance, and will be a useful addition to the radio propagation literature. I do have a few comments that should be considered by the authors before publication. COMMENTS 1. Although the authors clearly state that the importance of accurate Tmr depends on lower elevation angles, it is not clear to this reviewer if the analysis only uses Tmr at zenith. If so, please state explicitly what range of angles was used. It could be an interesting study of the dependence of the accuracy on elevation. 2. It was also not clear, what range of cloud liquid was encountered during

both the preparation and validation portion of the paper. Since the attenuation at V/W will be much more sensitive to cloud liquid than K band, perhaps some comments could be made on this point. Again, the dependence of Tmr errors on attenuation during cloudy conditions might be another useful study. The two channels at 51.26 and 52.26 GHz could prove useful in this situation. GRAMMATICAL COMENTS 1. P2,L51. Suggest replacing "band" by "bands". 2. Many pages, after defining PTU as pressure, temperature, and humidity, don't need to repeat. Also, for the first time, make it clear that relative humidity is used as the humidity variable. 3. P5,L 55. Please insert "a half data set". 4. P10, L19. Suggest "bring significant improvements" by deleting "to".

Please also note the supplement to this comment:
https://amt.copernicus.org/preprints/amt-2020-309/amt-2020-309-RC1-supplement.pdf

---

## Editor Comment (EC1) · Paolo Di Girolamo (Editor) · 26 Jan 2021

The short comment from the reviewer has to be handled as a referee comment.

---

## Author Comment (AC3) · 19 Feb 2021

Thank you very much for the suggestion, which we have pleasantly followed.

---

## Author Response (AR1)

**Reply to Referee Comment 1 (Ed R. Westwater)**

The paper on improving attenuation estimates by microwave radiometry is clearly written, of scientific significance, and will be a useful addition to the radio propagation literature. I do have a few comments that should be considered by the authors before publication.

Thanks for the positive feedback.

COMMENTS

1. Although the authors clearly state that the importance of accurate $T_{mr}$ depends on lower elevation angles, it is not clear to this reviewer if the analysis only uses $T_{mr}$ at zenith. If so, please state explicitly what range of angles was used. It could be an interesting study of the dependence of the accuracy on elevation.

Thanks for pointing this out. The whole analysis is performed at ~35° elevation, as the instruments are deployed within a radiopropagation experiment and thus constantly point towards the geostationary satellite Alphasat. We have added this information in Section 3.

2. It was also not clear, what range of cloud liquid was encountered during both the preparation and validation portion of the paper. Since the attenuation at V/W will be much more sensitive to cloud liquid than K band, perhaps some comments could be made on this point. Again, the dependence of $T_{mr}$ errors on attenuation during cloudy conditions might be another useful study. The two channels at 51.26 and 52.26 GHz could prove useful in this situation.

Thanks for pointing this out. Considering the pointing angle (35° elevation), cloud liquid water path reaches 2.8 (2.7) mm in the training (test) dataset. From the validation dataset, the estimated liquid water path reached 4.6 mm. We have added this information in Section 4. Thanks also for providing thoughts for further studies; we will keep into consideration as we go along the project.

GRAMATICAL COMMENTS

1. P2,L51. Suggest replace "band" by "bands".
2. Many pages, after defining PTU as pressure, temperature, and humidity, don't need to repeat.
Also, for the first time, make it clear that relative humidity is used as the humidity variable.
3. P5,L 55. Please insert "a half data set".
4. P10, L19. Suggest "bring significant improvements" by deleting "to".

Agreed. All the above suggestions are accepted and imported in the revised manuscript. Thanks much!

**Reply to Short Comment 1 (by Swaroop Sahoo)**

The paper is well written and explained. The analyses justifying the results are properly mentioned and explained.

Thanks for the positive feedback.

However, Appendix A needs to be written more comprehensively. Now it is just a bare minimum explanation. Also the mathematical formulation of the equation used in the appendix are not that clear. Readers will have difficulty in understanding the equations and the way they are written. The relationship between TMR and D is not very clear. This needs to be addressed.

Thanks for pointing this out. Appendix A has been completely rewritten in order to make it easier to understand and possibly reproduce the results. The relationship between $T_{MR}$ and D is now made explicit.

**Residual comments from initial review (anonymous Referee 2)**

Appendix A can be part of the main theme because it is not that long of a paper.

Appendix A was completely rewritten for sake of clarity. However, we prefer to keep it as stand-alone section, as it makes easier to locate the information needed to reproduce the results.

Time variation in attenuation should be presented because the title of the paper is based on attenuation not on mean radiating temperature.

More analyses on path attenuation has to be presented for the paper to be complete.

We added one figure (Figure 6 of revised manuscript) presenting 24-h time variation in attenuation, providing explanation for the discrepancies found between the conventional and proposed method.